# Mouse-Specific Single cell cytokine activity prediction and Estimation (MouSSE)

Azka Javaid *, H. Robert Frost

Department of Biomedical Data Science, Dartmouth College, Hanover, New Hampshire, United States of America

* azka.javaid@dartmouth.edu

**Data availability statement:** We provide the implementation of the MouSSE method, including the installation instructions and the associated vignette, via the MouSSE R package at https://github.com/azkajavaid/MousseR-package.

## Abstract

The accurate cell-level characterization of cytokine activity is important for understanding the signaling processes underpinning a wide range of immune-mediated conditions such as auto-immune disease, cancer and response to infection. We previously proposed the SCAPE (Single cell transcriptomics-level Cytokine Activity Prediction and Estimation) method to address the challenges associated with cytokine activity estimation in human single cell RNA-sequencing (scRNA-seq) and spatial transcriptomics (ST) data. Here, we propose a new method MouSSE (Mouse-Specific Single cell transcriptomics level cytokine activity prediction and Estimation) for performing cytokine activity estimation in murine scRNA-seq and ST data. MouSSE estimates the cell-level activity of 86 distinct cytokines using a gene set scoring approach. The cytokine-specific gene sets used by MouSSE are constructed using experimental cytokine stimulation data from the Immune Dictionary and cell-level scores are computed using a modification of the Variance-adjusted Mahalanobis (VAM) technique that supports both positive and negative gene weights. MouSSE is validated using data from both the Immune Dictionary via stratified cross-validation and external scRNA-seq and ST datasets against 10 cytokine activity estimation methods. These results demonstrate that MouSSE outperforms comparable methods for cell-level cytokine activity estimation in mouse scRNA-seq and ST data. An example vignette and installation instructions for the MouSSE R package are provided at https://github.com/azkajavaid/MousseR-package.

## Author summary

Herein, we present an overview of our recently developed cytokine activity estimation method, MouSSE (Mouse-Specific Single cell transcriptomics level cytokine activity prediction and Estimation). MouSSE estimates cell-level activity for 86 cytokines using gene sets constructed with cytokine stimulation data from the Immune Dictionary and scored with a modification of the Variance-adjusted Mahalanobis (VAM) method to support both positive and negative gene weights. We validate MouSSE against 10

**Funding:** This work was funded by the National Institutes of Health (R35GM146586 to HRF; R21CA253408 to HRF; P30CA023108 to HRF). The funders had no role in study design, data collection and analysis, decision to publish, or preparation of the manuscript. No authors received a salary from any of the funders.

**Competing interests:** The authors have declared that no competing interests exist.

different cytokine activity estimation techniques using stratified 5-fold cross-validation on Immune Dictionary-based mouse lymph node data and against publicly available COVID19-specific scRNA-seq and mouse lymph node-specific ST data. We quantify the performance of MouSSE using 11 metrics, including the Area Under the Receiver Operating Characteristic Curve (AUC-ROC), and the Precision-Recall Area Under the Curve (PR-AUC), amongst others, and perform sensitivity analysis for gene set size to extensively validate the gene set construction step in MouSSE. Overall, we conclude that MouSSE results in more accurate and biologically interpretable cytokine activity estimates as compared to alternative methods.

## 1. Introduction

Cytokines are secreted proteins that are produced by a range of immune and non-immune cells to regulate immune responses [1]. Pro-inflammatory cytokines, including interleukins such IL1$\beta$ and IL6, are important for initiating immune response to infections by promoting immune cell recruitment and activation, which is critical for fighting off infections. In comparison, anti-inflammatory cytokines, such as IL10, are essential in limiting inflammation and mitigating extensive tissue damage. These types of cytokines are known to promote tissue repair by promoting activation of regulatory immune cells [2]. Dysregulation of either pro-inflammatory or anti-inflammatory cytokines can result in impaired immune responses. For example, dysregulation of pro-inflammatory cytokines can trigger a cytokine storm, which is an excessive inflammatory response characterized by further production of cytokines with accompanying clinical symptoms including sepsis, hypotension and respiratory insufficiency [3]. Similarly excessive anti-inflammatory response can lead to immune suppression.

Given the potential of dysregulated cytokine activity to cause extensive damage, it is critical to effectively regulate cytokine levels. Effective regulation of cytokine activity can lead to the development of more targeted therapeutic interventions to control the often upregulated pro-inflammatory cytokine signatures, especially characteristic of conditions like cancer and autoimmune disorders like rheumatoid arthritis [4,5]. To effectively regulate cytokine activity, we need to develop methods to precisely quantify cytokine signaling activity levels. Since its introduction in 2009 [6], single cell RNA-sequencing (scRNA-seq) has enabled the transcriptomic profiling of tens to hundreds-of-thousands of cells from a single tissue sample [7]. Given its resolution, scRNA-seq technology can be leveraged to accurately quantify cytokine signaling at a single cell resolution. One challenge with estimating cytokine activity at the single cell-level is extreme sparsity (i.e., a large fraction of observed zero counts) and technical noise of scRNA-seq data [8,9]. Sparsity is especially problematic since it can result in reduced statistical power and can introduce artifacts in downstream analyses, including clustering and trajectory inference [10]. Despite the large number of scRNA-seq imputation methods [11], we found that the class of reduced rank reconstruction (RRR) methods, which assume that the intrinsic dimensionality of scRNA-seq data is much lower than the empirical rank, provide superior performance relative to other methods for scRNA-seq imputation and consequential reduction of sparsity. In light of this finding, we previously introduced methods SPECK (Surface Protein abundance Estimation using CKmeans-based clustered thresholding) [12] and STREAK (gene Set Testing-based Receptor abundance Estimation using Adjusted distances and cKmeans thresholding) [13] leveraging singular value decomposition (SVD)-based RRR for performing unsupervised and supervised receptor abundance

approximation, respectively, to generate non-sparse estimates of receptor transcripts. In addition to the application of RRR-based dimensionality reduction approaches to achieve a non-sparse and low-rank representation of gene expression data, gene set testing or pathway analysis approaches, that enable scRNA-seq downstream processing tasks to be performed on a pathway-level rather than on a gene-level, can be leveraged to mitigate sparsity and technical noise associated with scRNA-seq data analysis, thereby improving replication and interpretability [14,15]. For example, we developed the Variance-adjusted Mahalanobis (VAM) [16] method, a modification of the standard Mahalanobis distance that facilitates generation of cell-specific pathway scores, thereby accounting for the inflated noise and sparsity of scRNA-seq data. A related challenge is the often functionally redundant and pleiotropic signature of cytokines (i.e., overlapping activity of cytokines) [17]. For example, IL4 and IL13 share a common surface receptor to signal, thereby eliciting highly overlapping responses [18]. Given this functional redundancy, distinct estimation of cytokine activity is important for accurate signaling characterization.

In addition to developing methods to quantify cytokine signaling activity at a single cell resolution, we need to develop cell signaling methods that can accurately infer interaction activity (i.e., the receptor is present and bound by the cognate ligand) as compared to interaction potential (i.e., the receptor is present on the cell surface). The importance of developing methods that can distinctly infer interaction activity relates to a broader weakness of existing cell signaling characterization methods that make the assumption that gene expression alone reflects protein abundance, which reflects the strength of protein-protein interactions. Interaction activity, in comparison, is influenced by expression of downstream transcription and binding factors and therefore requires a more nuanced understanding of all interacting agents [19]. Even methods that account for downstream gene expression in estimating cell signaling, such as NicheNet [20], are limited since they depend on a prior definition of receiver and sender subpopulation, a characterization that is often challenged by the ambiguity of the location of the originating ligand given that the ligand may simultaneously originate from different groups of cells. Current cell signaling estimation methods such as CellChat [21], CellPhoneDB [22], SingleCellSignalR [23] and NicheNet are also limited in that they generate cell-cell communication or cell signaling over population clusters/cell-types. These methods do not generate cell signaling estimates at a single cell-level, thereby ignoring considerable within cluster heterogeneity. While methods that infer cell signaling at a single cell resolution by using deep learning to project the gene expression of single cells into a latent space, like SPRUCE [24] and DeepCOLOR [25], do exist [26], they are limited by transparency of their underlying computational approaches. In addition, many deep learning-based algorithms are not considerably biologically interpretable enough to be manually fine-tuned by medical practitioners [27].

Single cell signaling activity methods that leverage perturbation-based gene signatures are limited to a few pathways. PROGENy [28], for example, only supports 14 signaling pathways. We previously introduced our SCAPE (Single cell transcriptomics-level Cytokine Activity Prediction and Estimation) [29] method for cell-level cytokine activity estimation of human scRNA-seq data with support for 41 cytokines. While SCAPE works well in practice, it has a few practical limitations: 1) the gene sets used by SCAPE are based on bulk gene expression data from the CytoSig database [30] so may not match the equivalent signatures from scRNA-seq data and cannot provide cell type specificity, 2) SCAPE only supports a limited group of 41 cytokines, and 3) SCAPE only directly supports the analysis of human data.

To address these limitations and support cytokine activity estimation for murine scRNA-seq and ST data, we developed the MouSSE (Mouse-Specific Single cell transcriptomics level cytokine activity prediction and Estimation) method. MouSSE supports activity estimation

for 86 cytokines using gene sets constructed with scRNA-seq cytokine simulation data from the Immune Dictionary [31], which was recently developed to capture the *in vivo* transcriptomic impact of 86 different cytokines on individual immune cells extracted from mouse lymph nodes and profiled using scRNA-seq. Similar to the SCAPE technique, the MouSSE method computes cytokine-specific gene set signatures using differential expression analysis. In comparison to the SCAPE method, MouSSE specifically performs gene set scoring using a modification of the VAM technique that supports both positive and negative gene weights and is further adjusted for the number of genes falling into the positive and negative categories. Importantly, the MouSSE method's differential expression analysis uses scRNA-seq data from the Immune Dictionary. In proposing our MouSSE method, we aim to leverage the Immune Dictionary to generate perturbation-based signatures for each of the 86 cytokines and score these gene sets to generate single cell resolution measurements of cytokine activity.

## 2. Methods

### 2.1. Ethics statement

This study leveraged publicly available, de-identified COVID19 patient data originally published in Liao et al. [32] as part of the validation analyses. The original study obtained ethical approval from the relevant Institutional Review Boards and secured informed consent from all participants, as described in their publication. As this analysis used only publicly available, de-identified data, no additional Institutional Review Board approval or informed consent was required.

### 2.2. MouSSE method overview

MouSSE uses a gene set scoring approach to compute cell/location-level estimates of cytokine activity for murine scRNA-seq or ST data. Fig 1 provides a high-level visualization of the MouSSE technique. The gene sets corresponding to each of the 86 supported cytokines are generated through a differential expression analysis of the cytokine stimulation data from the Immune Dictionary, as detailed in Sect 2.4. Scoring of these cytokine-specific gene sets for target scRNA-seq/ST data is performed using a modification of the VAM method that supports both positive and negative gene weights and adjusts for the number of genes falling

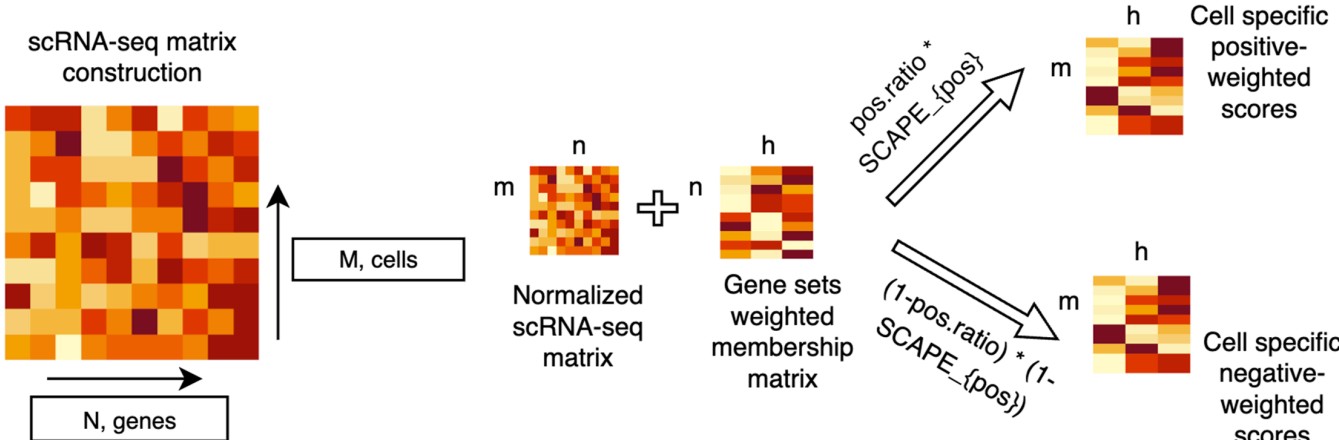

**Fig 1. MouSSE (Mouse-Specific Single cell transcriptomics level cytokine activity prediction and Estimation) supports both positive and negative gene set scoring with the modified VAM method constructed using each of the 86 cytokine-specific gene sets.**

into the positive and negative categories. This gene set scoring procedure is detailed in Sect 2.5.

## 2.3. Immune dictionary single cell data

To generate large-scale perturbation scRNA-seq data for 86 murine cytokines, the Immune Dictionary team injected a solution containing either a target cytokine or phosphate buffered saline (PBS) under the skin of C57BL/6 mice. Four hours after injection, skin-draining lymph nodes were collected. Following collection, cells were profiled using the 10x Chromimum system to generate scRNA-seq data for 386,703 cells. Demultiplexing and alignment of the sequencing data was processed using CellRanger v.3.0. Additionally, cell hashing was used to combine multiple scRNA-seq samples in the same channel [31].

We analyzed the processed scRNA-seq data generated by the Immune Dictionary project by matching gene expression and hashtag using the *HTODemux* function from the Seurat R package with the default positive.quantile probability of 0.99. We matched the quality control process from Cui et al. [31] by removing multiplets from our analysis and retaining cells with non-zero abundance values for more than 500 genes, more than 1,000 unique molecular identifiers and less than 10% mitochondrial gene content. Following this quality control, we preserved 385,174 cells for analysis.

## 2.4. Gene set construction

To create cytokine-specific gene sets, we performed differential expression analysis on the Immune Dictionary scRNA-seq data using a Wilcoxon rank sum test, as implemented by the *FindMarkers* function in the Seurat R package v5.0.0 [33]. Specifically, we compared the expression of each gene in cells extracted from mice that had been stimulated with a given cytokine against the expression of that gene in cells from mice stimulated with all other cytokines. For each cytokine, this analysis identified genes whose expression is either up or down-regulated in lymphatic tissue in response to *in vivo* cytokine stimulation. Importantly, comparing the stimulation results for each cytokine against all other cytokines and the control condition allowed us to capture differentially expressed markers that were specific to each cytokine relative to other potentially similar cytokines. This is in contrast to the differential expression test that is currently implemented in the Immune Dictionary-based Immune Response Enrichment Analysis (IREA), a companion software that compares each cytokine stimulation condition against only the PBS control to generate cytokine signatures in transcriptomes.

For each cytokine, we then selected the top 60 genes ranked by the absolute log2 fold-change in mean expression that were also statistically significant at an alpha level of 0.05. These genes were then separated according to the sign of the log2 fold-change into a positive set of size $n_{pos}$ (contains genes with log2 fold-change values above 0) and a negative set of size $n_{neg}$ (contains genes with log2 fold-change values below 0) with gene weights for both sets defined as the absolute value of the log2 fold-change. The final result of this procedure was a collection of 172 weighted gene sets comprising the positive and negative sets for each of the 86 supported cytokines.

## 2.5. Gene set scoring

To compute cytokine activity estimates for a target scRNA-seq or ST data set, MouSSE performs cell-level gene set scoring. For this task, we first compute cell/location-level scores for both the positive and negative gene sets for each cytokine with a modified version of the VAM method.

Implementation of the modified VAM method requires two input matrices:

1. $\mathbf{X}_R^*$: a $m \times n$ scRNA-seq matrix containing the normalized counts for $n$ genes in $m$ cells.
2. $\mathbf{A}$: a $n \times h$ matrix that captures the weighted annotations of $n$ genes to $h$ cytokine-specific gene sets.

VAM outputs matrix $\mathbf{S}$, that holds the cell-level scores for $m$ cells and $h$ gene sets. Below we detail the computation of $\mathbf{S}$ and of the matrix $\mathbf{M}$, which holds the cell-specific squared modified Mahalanobis distances for $m$ cells and $h$ gene sets.

1. **Technical variance estimation:** We first leverage Seurat's variance decomposition for log-normalized data to calculate the length $n$ vector $\sigma_{tech}^2$ holding the technical variance of each gene in $\mathbf{X}_R^*$.
2. **Modified Mahalanobis distances computation:** We next compute the cell-level squared distances for a column $k$ of $\mathbf{M}$, a $m \times h$ matrix of squared values of a modified Mahalanobis distance, as $M[,k] = diag(X_k(I_g\sigma_{g,tech}^2)^{-1}X_k^T)$. Here, $g$ corresponds to the gene set ($k$) size, $X_k$ is a matrix of size $m \times g$ which maps $g$ columns of $\mathbf{X}_R^*$ to the gene set size $k$, $I_g$ is a $g \times g$ identity matrix, and $\sigma_{g,tech}^2$ maps elements of $\sigma_{tech}^2$ associated to the $g$ genes in set $k$.
3. **Gamma distribution fit:** We next fit a gamma distribution to the non-zero elements in each column of $\mathbf{M_p}$ using the method of maximum likelihood.
4. **Cell-specific scores computation:** Lastly, we define the cell-level gene set scores, which are held in matrix $\mathbf{S}$, to be the gamma cumulative distribution function (CDF) value for each element of $\mathbf{M}$.

The modified VAM method is executed separately for the positive and negative weighted gene sets. The overall cell/location-level activity score of each cytokine, $s$, is then set to the weighted average of the VAM score for the positive set ($VAM_{pos}$) and 1 minus the VAM score for the negative set ($VAM_{neg}$) as defined by Eq 1.

$$s = \frac{n_{pos}}{n_{pos} + n_{neg}}\text{VAM}_{pos} + \frac{n_{neg}}{n_{pos} + n_{neg}}\left(1 - \text{VAM}_{neg}\right) \tag{1}$$

We note that the modified Mahalanobis distance implementation in VAM has two important differences from the standard Mahalanobis distance implementation. First, while the standard Mahalanobis distance leverages the full sample covariance matrix for estimation, the modified Mahalanobis distance only accounts for the technical variance of each gene, thereby ignoring covariances. Practically, this implies that VAM only discounts deviations in directions of large estimated technical variance while preserving deviations in directions of large biological variance (i.e., covariance). Second, while the standard Mahalanobis distance calculates distances from the multivariate mean, the modified Mahalanobis distance calculates distances from the origin, thereby resulting in a more biologically plausible distance measure for scRNA-seq data.

## 3. Evaluation

### 3.1. Benchmark setup and datasets

We performed stratified 5-fold cross-validation on the demultiplexed and pooled scRNA-seq data for each cytokine stimulation condition from the Immune Dictionary. Since this

scRNA-seq data contained 385,174 cells, our training data for each fold consisted of 308,141 cells and the test data consisted of 77,033 cells.

We evaluated MouSSE and comparative methods on scRNA-seq data generated from the Immune Dictionary using stratified 5-fold cross-validation. We further evaluated all methods on an additional scRNA-seq and an ST dataset. These datasets included: 1) Liao et al. [32] scRNA-seq dataset characterizing bronchoalveolar lavage fluid (BALF) immune cells from healthy individuals and patients with moderate and severe COVID19 and 2) Lopez et al. [34] ST dataset characterizing mouse lymph nodes treated with mycobacterium or PBS.

### 3.2. Comparative methods

We evaluated MouSSE against 10 different ligand-receptor interaction scoring strategies for scRNA-seq data as detailed below. The comparison methods included both cell-level signaling characterization methods and gene set-based techniques. We also assessed the efficacy of MouSSE by comparing it to MouSSE scored separately using the positive and negative weighted gene sets (i.e., comparison against all cytokines) and against gene sets constructed using the IREA method (i.e., comparison against PBS) but scored using the MouSSE method (i.e., weighting both positive and negative genes). We detail the implementation of each method below.

- **MouSSE positive-weighted (mousse.pos)**: MouSSE was scored using positive weighted gene sets (i.e., $\text{VAM}_{pos}$). Comparison of cytokine activity estimates generated using gene sets containing both positive and negative weights and those containing only positive weights enabled us to evaluate the effectiveness of the negative weighting mechanism as implemented in the MouSSE method.
- **MouSSE negative-weighted (mousse.neg)**: MouSSE was scored using negative weighted gene sets (i.e., $1 - \text{VAM}_{neg}$).
- **Normalized Ligand Score (naive)**: Normalized expression of the ligand gene transcript corresponding to each cytokine was set as a proxy for cytokine activity. For this purpose, we matched the genes corresponding to normalized gene expression data to each of the unique 86 cytokine treatment labels in order to extract the gene expression equivalent to each cytokine label. We obtained matching gene expression transcripts for 42 cytokines. We note that while this approach may not be practical in implementation, we used it for illustrative purposes.
- **Normalized Receptor Score (receptor)**: Normalized expression of the receptor(s) transcript corresponding to each cytokine was set as a proxy for cytokine activity. For this task, we referenced a mouse-specific ligand to receptor mapping database [35] that contains data for approximately 2,356 interactions. Approximately 40 out of the 86 cytokines contained matching receptors in this database. For 33 of the 40 cytokines, more than one receptor transcript matched the corresponding ligand. For these cytokines, we averaged the expression of the normalized receptor transcripts.
- **Ligand-Receptor Product Score (product)**: Product of the average ligand and receptor expression scores was set as a proxy for cytokine activity. The product score method was inspired by cell-cell communication methods that use the ligand-receptor expression product to quantify cell signaling at the cell level [36]. For this purpose, we multiplied the normalized expression of the cytokine with the normalized expression of the corresponding receptor(s) using the ligand-to-receptor mapping detailed in Cain et al. [35]. For cytokines with more than one receptor, we averaged expression across receptor transcripts.

- **NICHES (niches)**: NICHES [37] method was used to generate cell-level signaling characterization for cytokine activity. Since NICHES requires cell type information, we performed clustering using the Leiden algorithm of target scRNA-seq data with Seurat's default processing pipeline. NICHES also requires a ligand-to-receptor mapping. For this task, we leveraged the mouse-specific ligand-to-receptor mapping database from Cain et al. [35]. We then scored these 40 interactions using the *RunNICHES* function from the NICHES package v1.1.0 with the CellToSystem argument set to true to analyze cell-cell interactions without consideration of spatial coordinates on the normalized scRNA-seq data.
- **Seurat Perturbation Scores (seurat)**: Cell signaling estimates generated using the pathway-specific gene sets constructed by researchers from the Seurat lab for IFN$\beta$, IFN$\gamma$, TGF$\beta$ and TNF$\alpha$ pathways [38] were set as estimates for cytokine activity. For this task, we leveraged the up-regulated genes for each of the four pathways corresponding to the first program. We then averaged the expression for all genes in each of the four gene sets to compute cell-level signaling estimates.
- **PROGENy (progeny)**: Cell-level signaling estimates generated using the PROGENy method [39] were set as estimates for cytokine activity. While PROGENy can be used to infer activities for 14 signaling pathways, only five of the pathways overlapped with the 86 cytokines supported by MouSSE. These overlapping cytokines included EGFR, VEGF, TGF$\beta$, TNF$\alpha$ and Trail. We leveraged the *progeny* function from the progeny package v1.20.0 to infer pathway activities for these five cytokines.
- **IREA Positive-Weighted Scores (irea.pos)**: Cell signaling profiles generated using the IREA constructed gene sets representing the up and down-regulated gene signatures for each of the 86 cytokines against the PBS control were set as estimates for cytokine activity. Since the authors' implementation of the IREA software is not publicly accessible as an executable tool, we performed a comparative analysis using the gene sets provided in the Supplementary Table 3 of the original manuscript [31]. We first merged data from all gene sets and then selected the top 30 genes ranked by positive average log-fold change for each cytokine. We then averaged expression over all 30 genes to obtain an estimation of cytokine activity.
- **IREA Positive and Negative-Weighted Scores (irea.mousse)**: Cell signaling profiles generated using both the positive and negative weighted IREA constructed gene sets scored with the weighting strategy implemented in the MouSSE method were set as estimates for cytokine activity. This implementation of IREA using the weighting strategy proposed in the MouSSE method allowed us to determine the efficacy of both our positive and negative gene weighting methodology and our gene set construction strategy, which computes up and downregulated differentially expressed genes for each of the 86 cytokines compared to all other cytokines (i.e., against the PBS condition), as currently implemented in the IREA software.

### 3.3. Performance metrics

We evaluated MouSSE and comparative methods using 11 metrics, including the Area Under the Receiver Operating Characteristic Curve (AUC-ROC), Precision-Recall Area Under the Curve (PR-AUC), sensitivity, specificity, Negative Predicted Value (NPV), precision, F1 score, prevalence, detection rate, detection prevalence and balanced accuracy to quantify the correspondence between each method and the ability to correctly predict the cytokine label corresponding to each sample. We quantified the one-vs-rest AUC-ROC on the continuous cytokine activity estimates using the *auc* function from the pROC R package v1.18.5. To

compute the PR-AUC, we used the *pr.curve* function from the PRROC R package v1.4. To compute the remaining metrics, we constructed a Receiver Operating Characteristic (ROC) curve with the *roc* function and then determined the optimal threshold for each cytokine with the *coords* function from the pROC package. More precisely, we set the optimal threshold as the point closest to the top left part of the ROC plot with perfect sensitivity or specificity, as indicated by the optimality criterion-specific Eq 2 below. Lastly, we binarized our continuous cytokine activity prediction estimates using this threshold, assigning a value of 1 to predictions greater than the threshold (i.e., prediction >threshold = 1), and 0 otherwise. We then created a confusion matrix with the *confusionMatrix* function from the caret R package v7.0.1 and extracted the sensitivity, specificity, NPV, precision, F1 score, prevalence, detection rate, detection prevalence, and balanced accuracy metrics accordingly as defined by [40].

$$optimality\ criterion = \min\left((1 - sensitivities)^2 + (1 - specificities)^2\right) \tag{2}$$

## 4. Results

### 4.1. MouSSE generates accurate cytokine activity estimates

**4.1.1. Assessment using the Area Under the Receiver Operating Characteristic Curve (AUC-ROC).**  We first evaluated MouSSE against the comparative methods listed in Sect 3.2 on the Immune Dictionary-based mouse lymph node scRNA-seq data. For this evaluation, we quantified the proportion of cytokines with the highest AUC-ROC across every method. For our initial comparison, we compared MouSSE against other signaling estimation methods, including normalized ligand (naive), normalized receptor (receptor), ligand-receptor product (product), NICHES (niches) and PROGENy (progeny). As Fig 2A shows, 99% of cytokines have the highest AUC-ROC when estimated using the MouSSE method as compared to other signaling characterization methods. Next, we compared MouSSE against other gene set construction strategies, including IREA-based gene sets (irea.pos) and gene sets built using Seurat's perturbation scores (seurat). Fig 2B shows that approximately 91% of cytokines have the highest AUC-ROC when estimated using the MouSSE method as compared to the irea.pos and seurat methods.

To further assess the effectiveness of MouSSE's positive and negative gene set weighted scoring strategy, we compared the cell signaling estimates generated using MouSSE against estimates generated using each of the positive or negatively differentially expressed gene sets (i.e., mousse.neg and mousse.pos, respectively) and against gene sets constructed using IREA but scored for both positive and negative genes using the gene set scoring strategy proposed in the MouSSE method (irea.mousse). We note that comparing MouSSE against cytokine activity estimates generated using solely positive or negatively differentially expressed genes, as well as with gene sets constructed using the IREA method but scored using MouSSE's weighting approach, enabled us to assess the effectiveness of MouSSE's gene set construction and weighting algorithm. As shown in Fig 2C, approximately 75% of cytokines have the highest AUC-ROC score when estimated using the MouSSE method. In comparison, 18% of cytokines have the highest AUC-ROC score when estimated using irea.mousse and 5% of cytokines have the highest AUC-ROC score when estimated using the mousse.neg method.

Fig 2D visualizes the results from comparing MouSSE against all 10 cytokine activity estimation strategies. In this scenario, approximately 73% of cytokines have the highest AUC-ROC when estimated using MouSSE. By contrast, this proportion is only 13% for irea.mousse, and 5% for each of the irea.pos and mousse.neg methods, and 3% for mousse.pos.

In addition to reporting the proportion of cytokines that have the highest AUC-ROC when estimated using MouSSE and comparative methods, Fig 3 visualizes the individual

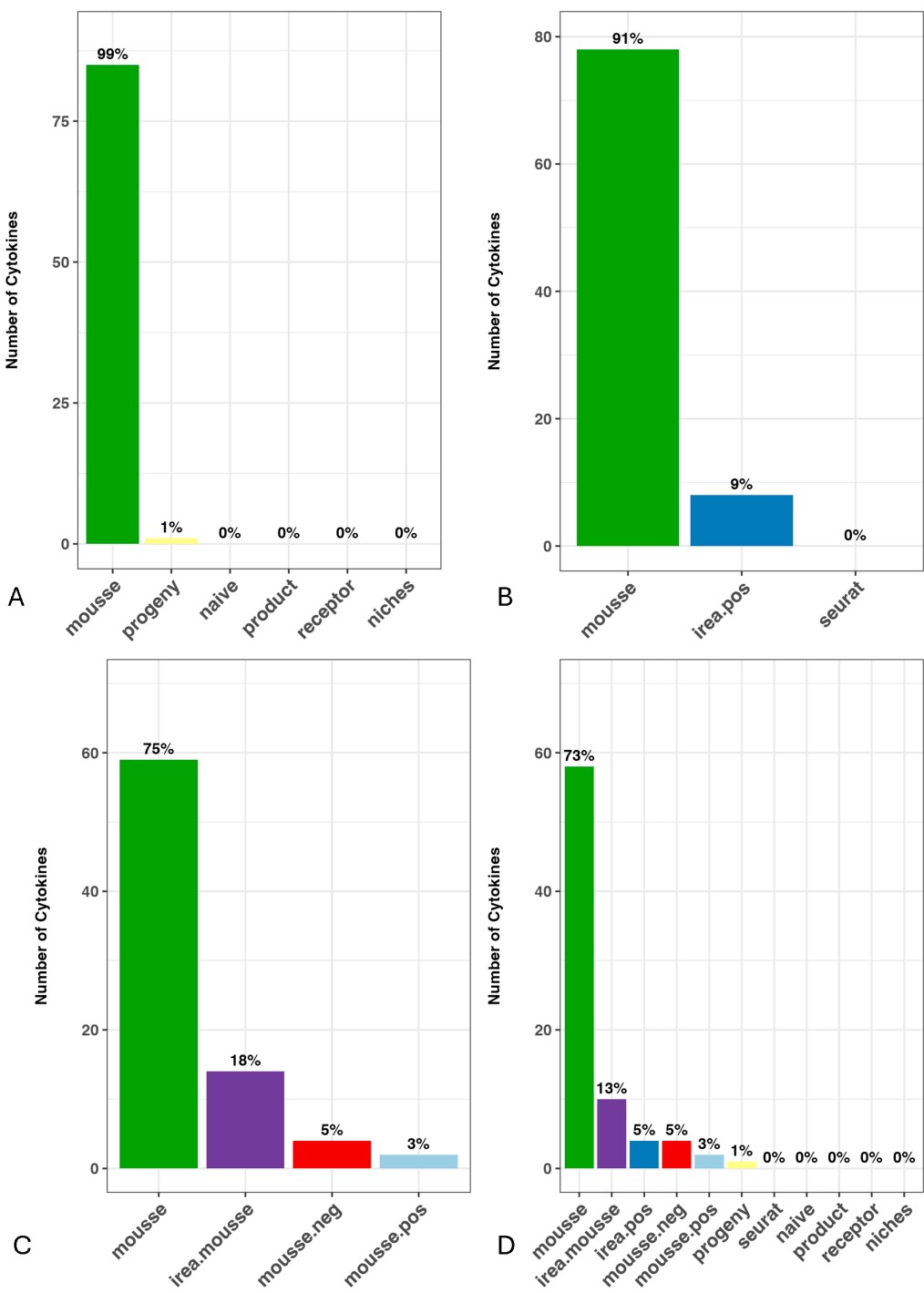

**Fig 2. Proportion of cytokines with the highest Area Under the Receiver Operating Characteristic Curve (AUC-ROC) score calculated from cross-validated mouse lymph node target scRNA-seq data, encompassing 77,033 cells.** (Fig 2A) Comparison of MouSSE scores (mousse) with other signaling estimation methods, including normalized ligand (naive) and receptor scores (receptor), ligand-receptor product (product), NICHES application (niches), and PROGENy application (progeny). (Fig 2B) Evaluation of MouSSE scores relative to alternative gene set construction approaches, such as IREA-based gene sets scored using only positively weighted genes (irea.pos) and Seurat-based perturbation scores (seurat). (Fig 2C) Sensitivity analysis of MouSSE's gene set construction and scoring methodology, comparing it with IREA-based gene sets incorporating both positive and negative weights and scored using MouSSE's weighting strategy (irea.mousse) and modified MouSSE approach using only positively weighted genes (mousse.pos) and using only negatively weighted genes (mousse.neg). (Fig 2D) Overall comparison of MouSSE scores across all ten methods.

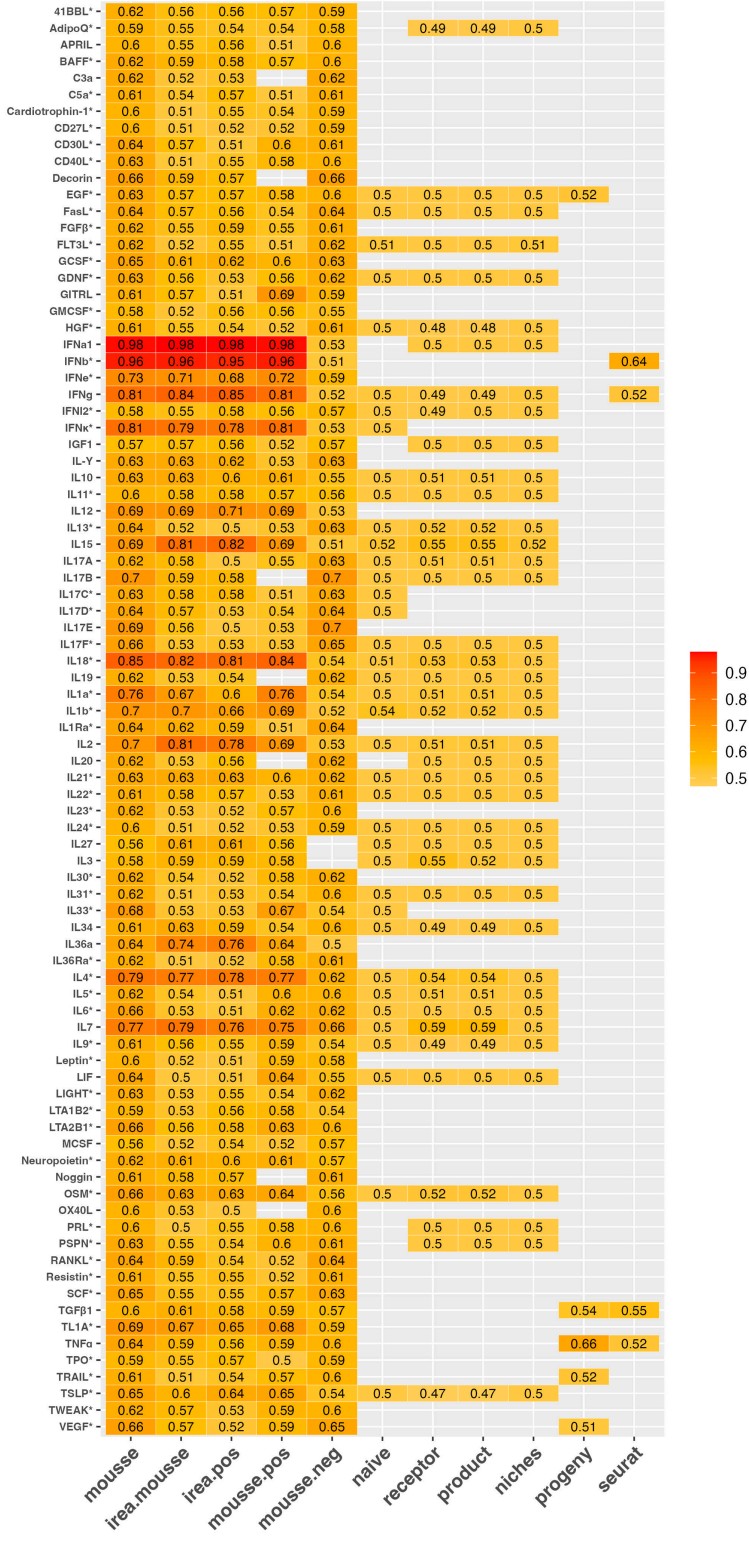

**Fig 3. Average Area Under the Receiver Operating Characteristic Curve (AUC-ROC) for all 86 cytokines quantified using the MouSSE and comparative methods as computed on the 77,033 cell mouse lymph node target scRNA-seq data.** Cytokine markers with an asterisk have the highest AUC-ROC score when estimated using the MouSSE method.

AUC-ROC scores for every cytokine for activity estimates generated using all comparative techniques. This heatmap indicates that while estimates for cytokines like IL18 have a much better correspondence with sample labels when estimated by MouSSE (AUC-ROC = 0.85), other cytokines like IL2 show better correspondence for estimates generated using alternative methods (i.e., AUC-ROC for irea.mousse = 0.81 versus AUC-ROC for MouSSE = 0.7). This heatmap can be referenced to identify the optimal activity estimation strategy for each profiled cytokine. Similar heatmaps documenting the individual PR-AUC, specificity, sensitivity, precision, NPV, prevalence, F1 score, detection rate, detection prevalence and balanced accuracy are available in S1–S10 Figs, respectively.

**4.1.2. Assessment using classification performance metrics.** We next evaluated MouSSE against comparative methods on the Immune Dictionary-based mouse lymph node scRNA-seq data using sensitivity, specificity, NPV, precision, F1 score, prevalence, detection rate, detection prevalence and balanced accuracy. Similar to the analysis above, we quantified the proportion of cytokines with the highest metric-specific score across every method. We report our findings in Fig 4.

Overall, we found that our MouSSE method has the largest percentage of cytokines with the highest balanced accuracy (59%), F1 score (48%), NPV (44%) and precision (41%). Given that our dataset contained a comparatively smaller number of positive cytokine labels, the F1

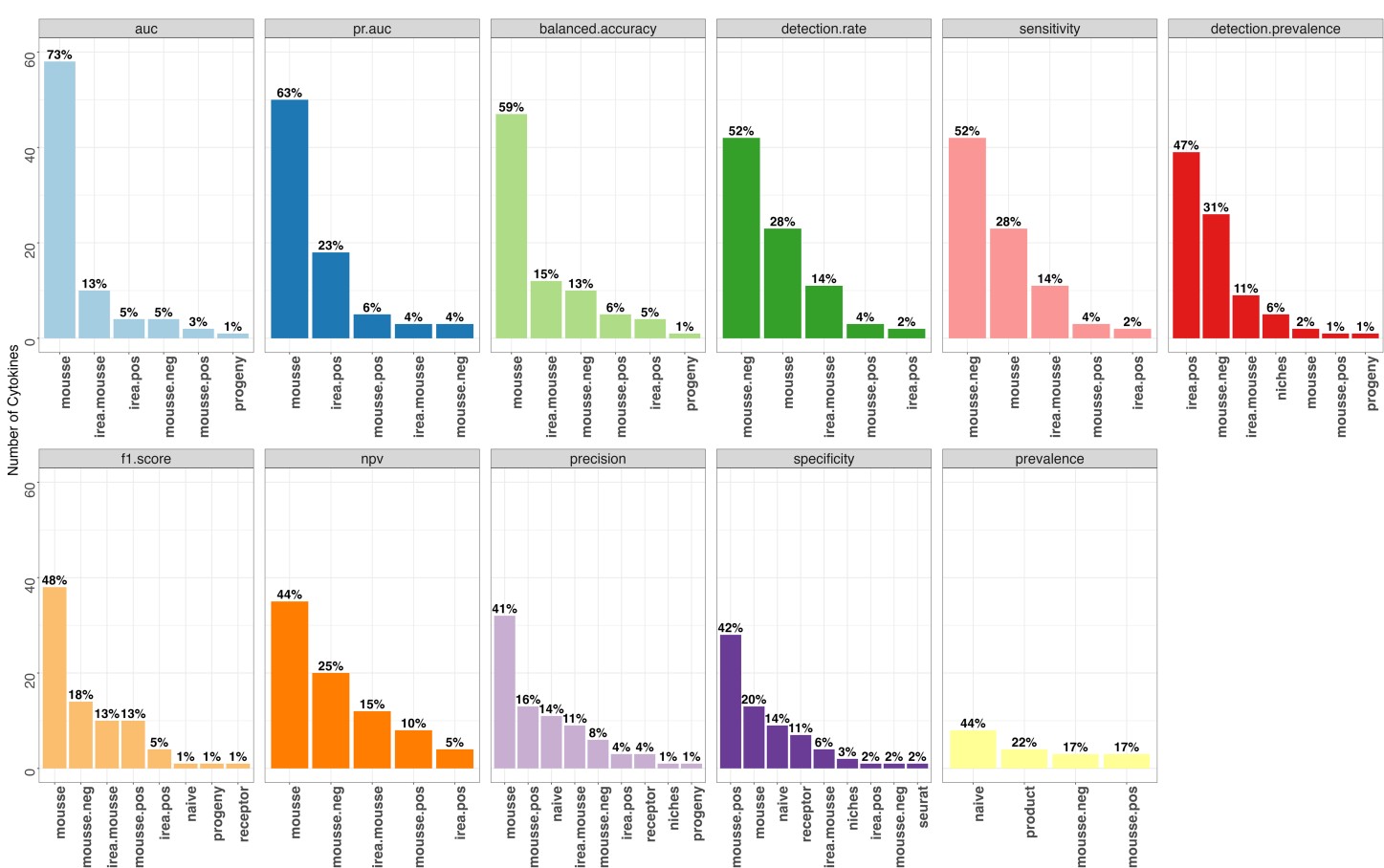

**Fig 4. Proportion of cytokines with the highest AUC-ROC, PR-AUC, balanced accuracy, detection rate, sensitivity, detection prevalence, F1 score, NPV, precision, specificity and prevalence across each of the 10 methods as computed from cross-validated mouse lymph node target scRNA-seq data, encompassing 77,033 cells.**

score, which is the harmonic mean of the precision and recall, allowed us to correctly evaluate the trade-off between false positives and false negatives. We report that approximately 48% of cytokines have the highest F1 score for cytokine activity estimates generated using the MouSSE method as compared to 18% of cytokines that have highest F1 score for activity estimates generated using the mousse.neg method and 13% of cytokines that have the highest F1 score for estimates generated using each of the irea.mousse and mousse.pos methods. While the F1 score allowed us to prioritize the correct detection of the positive class (i.e., active cytokine signature labels), we also computed the balanced accuracy to assess the performance of all methods with respect to both sensitivity and specificity, thereby giving equal weight to both false positives and false negatives. We report that approximately 59% of cytokines have the highest balanced accuracy for cytokine activity estimates generated using the MouSSE method as compared to 15% of cytokines that have the highest balanced accuracy for activity estimates generated using the irea.mousse method.

**4.1.3. Assessment using the Precision-Recall Area Under the Curve (PR-AUC).** To further account for the imbalanced nature of the cytokine classification task and the rarity of positive cytokine instances in our dataset, we computed the PR-AUC, which allowed us to examine the trade-off between precision and recall across different thresholds. As shown by Fig 4, we found that approximately 63% of cytokines have the highest PR-AUC score for cytokine activity estimates generated using the MouSSE method as compared to 23% for irea.pos and 6% for mousse.pos. While MouSSE has comparatively higher PR-AUC values than other methods, the individual PR-AUC scores are considerably small. We hypothesize that the small PR-AUC values likely result from the inability of MouSSE and comparative methods to generate a clearly distinct signature for each cytokine relative to all other cytokines. The challenge in distinguishing between different cytokines stems from both the fact that there are groups of cytokines with similar activity profiles and the nature of cytokine stimulation experiments used to generate the Immune Dictionary data. Specifically, the Immune Dictionary scRNA-seq data was generated on lymph nodes extracted four hours after distal injection of a specific cytokine. This delayed tissue collection means that the measured expression profile is the result of not just the injected cytokine but downstream signaling as well as signaling that would normally take place within the lymph node. To further analyze this result, we generated differential expression-based gene sets for each cytokine relative to every other cytokine. We then computed pairwise PR-AUC scores on the test data to assess MouSSE's ability to distinguish each cytokine's signature from every other cytokine. We report our findings in S11 Fig. We observe that the interferon-specific cytokines have high PR-AUC scores against almost all cytokines. Similarly, cytokines like IL18 and IL7 have relatively higher PR-AUC scores against most other cytokines. In addition to the heatmap reporting the individual PR-AUC scores for every pairwise cytokine comparison, we provide the heatmap reporting the individual AUC-ROC, specificity, sensitivity, precision, NPV, prevalence, F1 score, detection rate, detection prevalence and balanced accuracy for every pairwise cytokine comparison in S12–S21 Figs, respectively. These heatmaps can be referenced to identify pairs of cytokines that are readily distinguished based on the magnitude of each score.

**4.1.4. Sensitivity analysis of MouSSE to gene set size.** To examine the influence of gene set size on cytokine activity estimation, we compared the cytokine activity estimates generated by MouSSE using gene sets that include the top 20, 60, 100 or 200 genes by absolute avg_log2FC threshold. As shown in S22–S25 Figs for set sizes of 20, 60, 100 and 200 genes, respectively, the proportion of cytokine activity estimates that have each of the highest AUC-ROC, PR-AUC, balanced accuracy, F1 score, NPV and precision when scored using the

MouSSE method as compared to the rest of the methods generally increases with increasing gene set size with a considerable increase reported from gene set size consisting of 20 genes to 60 genes. We conclude that a gene set size greater than the default of 60 can potentially improve model performance. We provide support for customizing the set size parameter in our *mousse* function with the *numGenes* parameter in the mousseR R package v1.0.0.

To further analyze differences in gene set size against the avg_log2FC threshold, we visualized the minimum absolute avg_log2FC threshold for each of the 86 cytokines for gene set sizes of 20, 60, 100 and 200 genes as averaged over the five folds of cross-validation. This analysis allowed us to examine how variations in the selection of the minimum absolute avg_log2FC value used to obtain the trained gene sets might be linked to cytokine-specific performance differences in test data. We visualize these thresholds and the sum of the thresholds over all gene set sizes for every cytokine in Fig 5. We found that cytokines of the interleukin-1 (IL-1) family (i.e., IL1$\alpha$, IL1$\beta$, IL36$\alpha$, IL33 and IL18) have a comparatively high minimum absolute avg_log2FC threshold over all gene set sizes compared to cytokines of other families. In addition, we observed that interferons, especially IFN$\alpha$1 and IFN$\beta$, have the highest minimum absolute avg_log2FC threshold over all gene set sizes (sum of minimum absolute avg_log2FC over gene set size = 11.2 and 10.4 for IFN$\alpha$1 and IFN$\beta$, respectively). These higher average log-based fold-change values for cytokines specific to the IL1 and interferon families relative to the remaining cytokines indicate that cytokines of these families exhibit more distinctive expression signatures that are more readily distinguishable from other cytokines.

Lastly, we quantified the computational complexity of MouSSE and comparative methods like NICHES and PROGENy. We found that MouSSE is relatively more computationally expensive as it takes about 8.2 minutes to run, as averaged over the five folds of cross-validation. In comparison, NICHES takes about 7.0 minutes and PROGENy takes about 6.8 minutes. We chose not to calculate the computational complexity of the remaining methods as they are largely averaging expression over gene sets. Therefore, their computational complexity is fairly negligible in comparison to methods like MouSSE, NICHES and PROGENy. We observe that while MouSSE is computationally more expensive, its performance outweighs the performance of comparative activity estimation strategies.

## 4.2. Application of MouSSE to COVID19 scRNA-seq data

We next applied MouSSE and comparative methods to the Liao et al. [32] COVID19 dataset, which contains transcriptomic measurements for three control, three mild and six severe COVID19 patients. For this analysis, we aimed to examine whether existing cytokine and cell signaling characterization methods recapitulate the ground truth regarding known cytokines that may be up or downregulated by COVID19 severity. To perform quality control, we followed the original criteria outlined by the authors (i.e., gene number between 200 and 6,000, UMI count >1,000 and mitochondrial gene percentage <0.1). We next merged data across replicates and integrated across disease severity using Seurat's Canonical Correlation Analysis (CCA). Our merged dataset consisted of 63,734 samples and 2,000 features. We then performed cytokine activity estimation using MouSSE and comparative methods on this aggregated data. Following estimation, we scaled the generated cell signaling estimates and used the *FindAllMarkers* function from the Seurat package to identify the top 15 cytokine markers by avg_log2FC that are differentially expressed based on COVID19 severity condition (i.e., control, mild and severe). We did not generate cytokine activity estimates for the COVID19 dataset for the NICHES method since the *RunPCA* function, a requirement for NICHES, did not return any features with any variance.

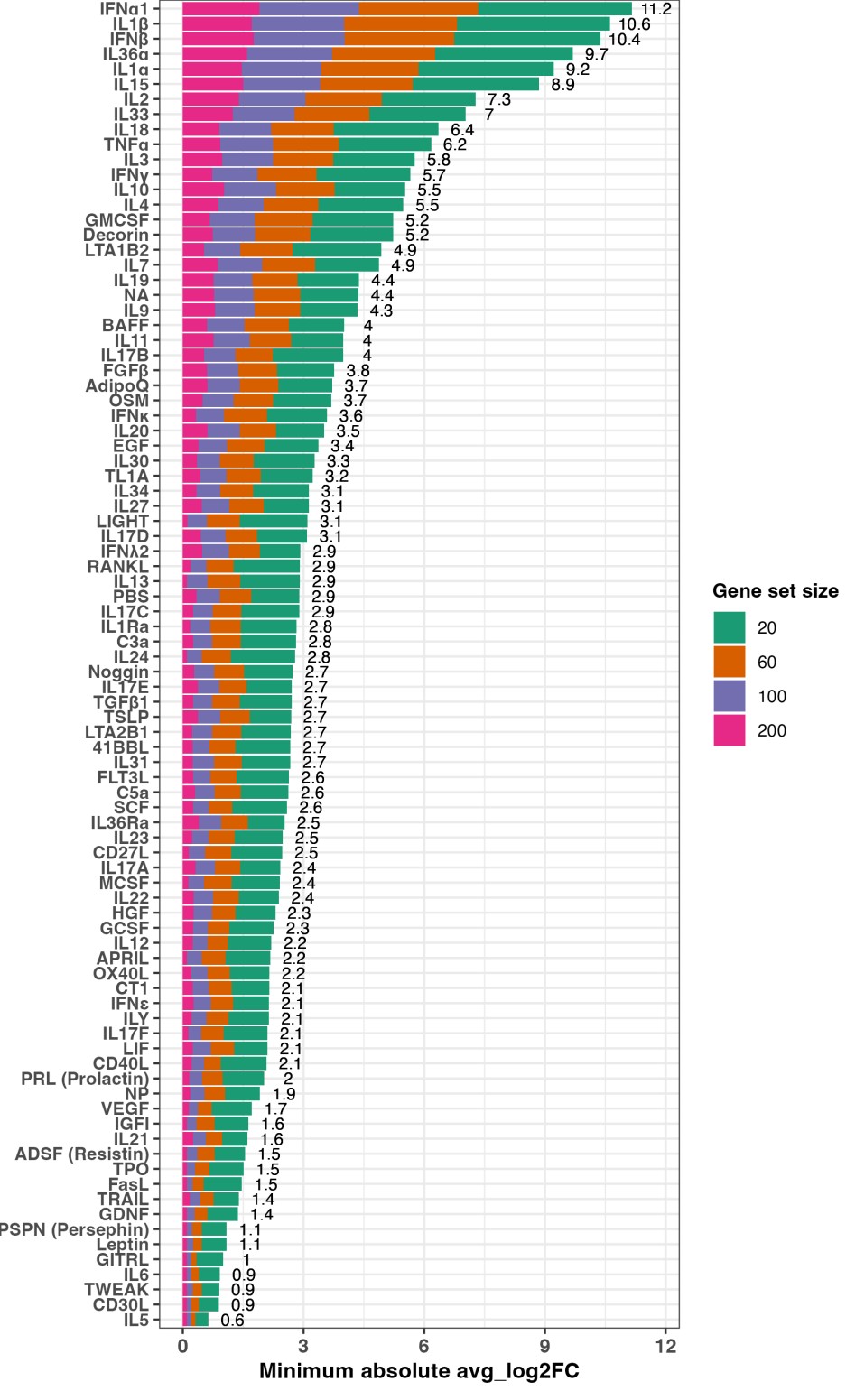

**Fig 5. Minimum absolute avg_log2FC threshold for each of the 86 cytokines for gene set size ranging from 20, 60, 100 and 200 genes as computed from cross-validated mouse lymph node target scRNA-seq data.** Sum of thresholds over all gene set size is indicated for each cytokine.

We first visualize the MouSSE-specific cytokine signaling estimates that are differentially expressed based on COVID19 severity. As Fig 6 shows, markers specific to interferon signaling (i.e., IFN$\alpha$1, IFN$\beta$, IFN$\gamma$) are elevated and differentially expressed in patients with severe COVID19 compared to patients with control and mild COVID19 for signaling estimates generated using the MouSSE method. While earlier studies reported impaired and limited response of type I interferons (IFN-I) (i.e., IFN$\alpha$1, IFN$\beta$) in response to COVID19, there is now increased evidence for robust IFN-I responses in patients with severe COVID19. One study reports that 15% of patients with critical COVID19 have auto-antibodies to interferon cytokines [41]. More precisely, an immune landscape study performed scRNA-seq analysis in peripheral blood mononuclear cells (PBMCs) in patients with mild or severe COVID19 and reported that IFN-I responses co-occurred with inflammatory responses driven by TFN- and IL1-driven cytokines in patients with severe COVID19 compared to patients with mild COVID19 [42]. This study supports our findings as we report the upregulation of IFN-I cytokines and upregulation of TNF$\alpha$, IL1$\alpha$ and IL1$\beta$ in severe COVID19 compared to control and mild COVID19. In addition to these results, we report the upregulation of IL6 in severe COVID19 as compared to control and mild COVID19. The upregulation of IL6 is supported by medical literature as increased IL6 levels are a critical mediator of hyperinflammation, which is observed to be considerably higher in severe COVID19 as compared to mild

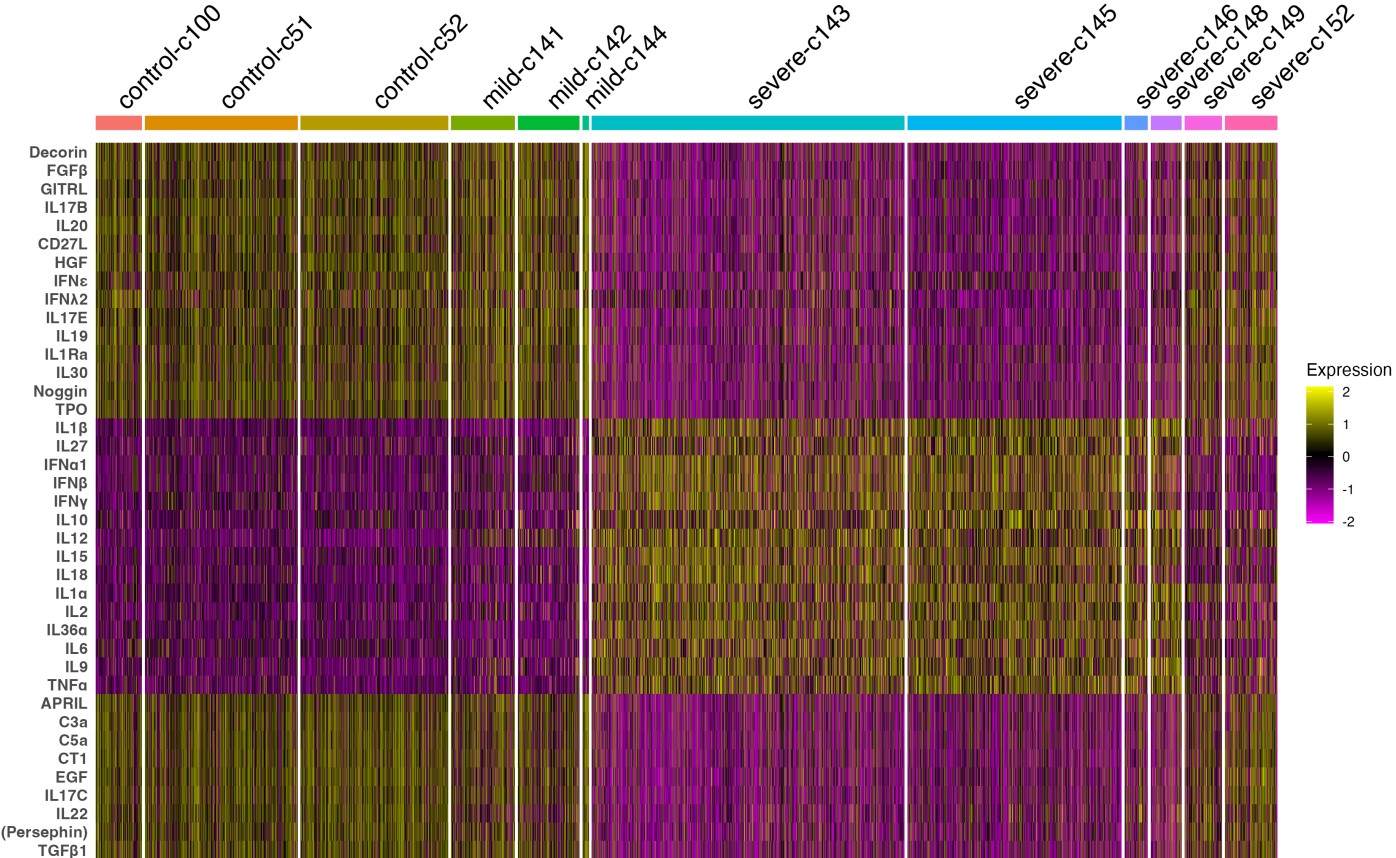

**Fig 6. Differentially expressed cytokine markers on scRNA-seq data consisting of 63,734 samples by COVID19 severity condition (i.e., control, mild and severe) for the MouSSE method.**

COVID19 patients [43]. We also show that the identified differentially expressed cytokines are mostly consistent between mild COVID19 and control patients. More specifically, we report that IFN$\epsilon$ is more distinctly elevated in the mild COVID19 condition compared to the control and severe COVID19 conditions. From literature, IFN$\epsilon$ is known to have weaker antiviral and natural killer cell cytotoxicity activities compared to the IFN-I cytokines [44,45]. IFN$\epsilon$'s upregulation in the mild COVID19 condition is therefore justified given its weaker inflammatory properties compared to IFN-I cytokines.

Our results also show some stratification of differentially expressed markers by different patient-specific samples within the severe COVID19 condition. For example, differentially expressed markers specific to the severe-c149 and severe-c152 patients show some overlap with differentially expressed markers identified for the mild COVID19 and control patients. On further inspection, we find that while patients c149 and c152 both had severe COVID19, they were relatively younger than their other severe counterparts. Patient c149 was 57 years old, for example, while patient c152 was 46 years old. In comparison, most other severe COVID19 patients were in their 60s (i.e., severe-c143, severe-c145, severe-c146 and severe-c148 patients were 62, 66, 63 and 65 years old, respectively). We find support for our assertion by Liao et al. [32] as they report that while patient c152 was previously in critical condition, they later improved.

We report that while the results generated by MouSSE support the expected upregulation of canonical markers of COVID19 by disease severity and the expected stratification within COVID19 severity categories, the results generated by comparative methods do not support a similar pattern of findings. For example, we find that while interferon-specific cytokines like IFN$\alpha$1 and IFN$\gamma$ are upregulated in patients with severe COVID19 for activity estimates generated using the mousse.pos (see S26 Fig) and mousse.neg (see S27 Fig) methods, there is not necessarily a strict, clustered separation between differentially expressed markers by disease severity. Similarly, there is not a strict upregulation of markers specific to severe COVID19 (i.e., IFN$\alpha$1, IFN$\beta$, IFN$\gamma$ and IL6) relative to control and mild COVID19 for differentially expressed markers generated using the cytokine-specific estimates produced by the irea.pos method (see S28 Fig). We do find however that the implementation of our gene set scoring strategy with the IREA-specific gene sets (i.e., irea.mousse) considerably improves the performance of the IREA method (see S29 Fig). More specifically, we find that interferon-specific cytokines like IFN$\alpha$1 and IFN$\beta$ are correctly upregulated in severe COVID19 as compared to control and mild COVID19. Additionally, these markers are not strictly upregulated in severe-c149 and severe-c152 patients, which is a result that is supported by our MouSSE method. We also find that IL6 is not necessarily upregulated in severe COVID19 as compared to control and mild COVID19 for cytokine activity estimates generated using the irea.mousse method. In this regards, MouSSE is relatively better and more specific than the cyokine activity estimates generated using the irea.mousse and comparative methods since MouSSE generates cytokine activity estimates that result in identification of IL6 as a distinct marker for severe COVID19. We also report that activity estimates generated using the naive (S30 Fig), receptor (S31 Fig), product (S32 Fig), Seurat (S33 Fig) and PROGENy methods (S34 Fig) are relatively noisy and less informative than other gene set-specific methods, thus failing to fully account for signaling variations by disease severity.

**4.2.1. Correlation of MouSSE-based cytokine activity estimates and MSigDB Hallmark gene sets.** To further validate the MouSSE method, we computed a correlation matrix between the cytokine activity estimates generated using MouSSE and the scored MSigDB Hallmark gene sets [46]. Since the hallmark signatures contain inflammation specific gene sets, we hypothesized a strong correlation between the IFN-I-specific cytokine activity

estimates generated using MouSSE and the inflammatory-specific hallmark gene sets. For this task, we retrieved all gene sets specific to the Hallmark collection using the *msigdbr* function from the msigdbr R package v10.0.1. We then scored each Hallmark gene set by averaging the expression of all genes in that set. We visualized the correlation matrix between the cytokine markers differentially expressed by COVID19 severity as generated using the MouSSE method and the scored Hallmark gene sets using a heatmap. As Fig 7 shows, the IFN-I-specific cytokines (i.e., IFNα1 and IFNβ) are highly correlated with the interferon alpha and interferon gamma gene sets. In addition, cytokines with differential expression patterns specific to severe COVID19 (i.e., TNFα, IL1α, IL1β and IFN-I) are all highly correlated with the inflammatory response gene set. In comparison, IFNε, which is more specific to mild COVID19 as indicated by Fig 6, has a relatively lower correlation with the inflammatory response-specific sets. These results indicate that the cytokine activity estimates generated using the MouSSE method, especially the IFN-I and pro-inflammatory-specific cytokine estimates, are correlated with the expected Hallmark pathways.

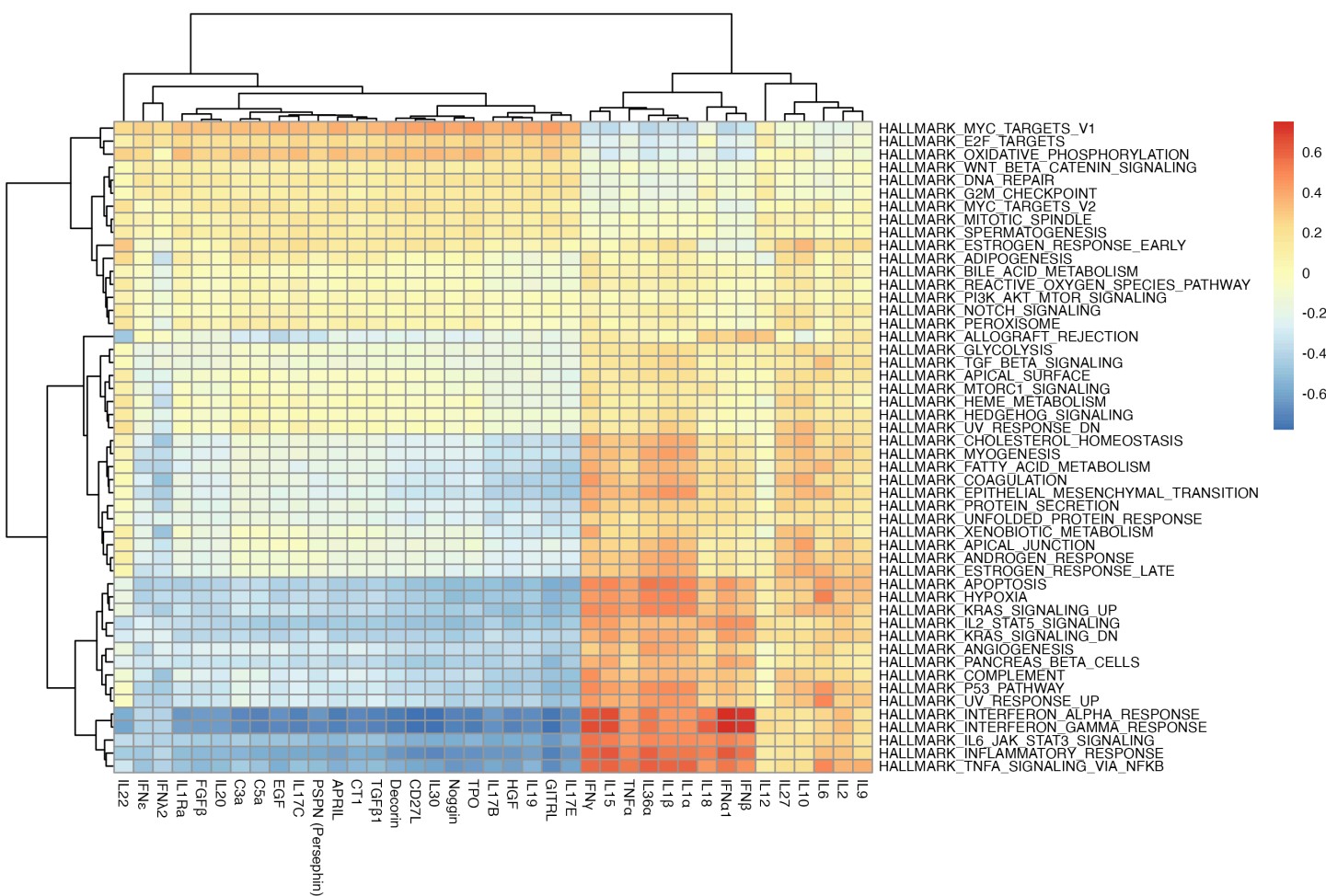

**Fig 7. Heatmap of Pearson correlation between cytokines differentially expressed by COVID19 severity as estimated using the MouSSE method (see Fig 6) and the Hallmark-specific pathway scores as computed on the COVID19 scRNA-seq data consisting of 63,734 samples.**

## 4.3. Application of mousse to mouse lymph node spatial transcriptomics data

We next applied MouSSE to the Lopez at el. mouse lymph node Visium spatial transcriptomics (ST) dataset, which examines lymph nodes following 48-hour stimulation with *Mycobacterium smegmatis* (MS). Since MS induces a response characterized by IFNγ [47], we expected to see upregulation of the interferon-specific response in the cytokines differentially expressed between the infected and control lymph nodes. For this task, we used the SpatialD-DLSdata R package v0.1.0 [48], which contains the prepackaged dataset consisting of 1092 spots. We log-normalized this ST data and applied the MouSSE method to generate cell signaling estimates. We then scaled these estimates and used the *FindAllMarkers* function from the Seurat package, with the minimum detection rate (min.pct) set to 0.25 using the ROC test, to find cytokines differentially expressed by infection status (i.e., MS-1/MS-2 infected and PBS control).

We first visualize the MouSSE-specific cytokine signaling estimates that are differentially expressed based on infection status. Fig 8A shows that the cytokines upregulated in the infected category relative to PBS include IFNκ, IFNε and Neuropoietin. In comparison, LIF (Leukemia inhibitory factor) is upregulated in the PBS condition relative to the infected category. Both IFNκ and IFNε are interferon-specific signaling agents while LIF is known to have anti-inflammatory and analgesic properties [49]. The upregulation of interferon-specific signaling agents in the infected categories (i.e., MS-1/MS-2) and the upregulation of anti-inflammatory-specific signaling agent in the control category (i.e., PBS) shows an accurate recapitulation of variations in signaling patterns based on the treatment condition as estimated using the MouSSE method.

Fig 8B demonstrates that the estimates generated using the mousse.pos method perform comparably to those generated by the MouSSE method. Both approaches identify similar interferon-specific cytokines (i.e., IFNε, IFNκ, and Neuropoietin) as upregulated in the infected categories relative to PBS, while the LIF cytokine is upregulated in the PBS condition compared to the infected categories, consistent with results from the original MouSSE method.

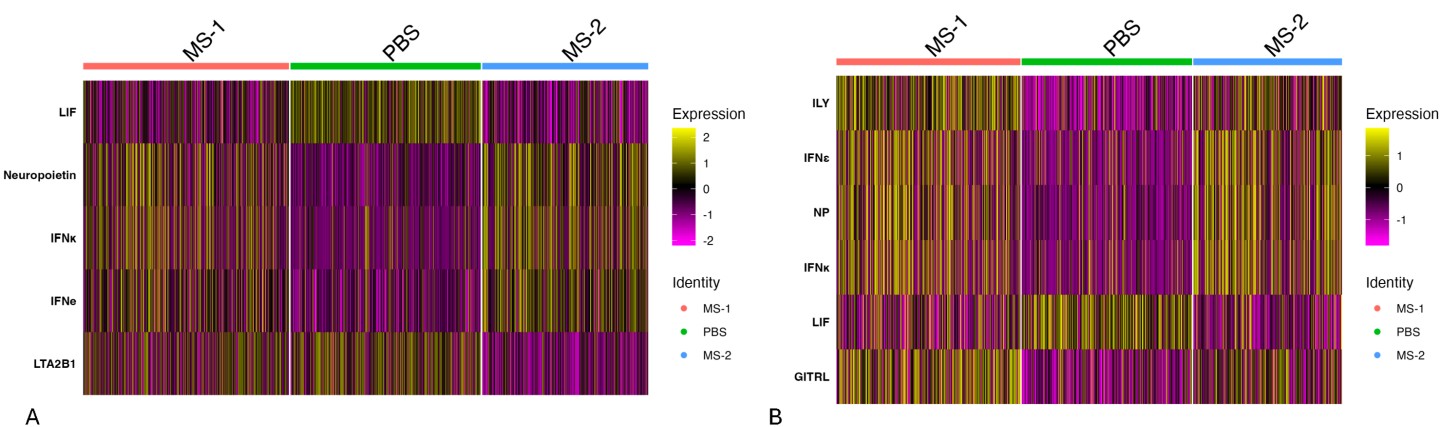

**Fig 8. Differentially expressed cytokine markers on mouse lymph node ST data consisting of 1092 spots by infection status (i.e., MS-1, MS-2 and PBS).** (Fig 8A) Heatmap of differentially expressed cytokines for the MouSSE method (mousse). (Fig 8B) Heatmap of differentially expressed cytokines for the modified MouSSE method scored on just the positive weighted genes (mousse.pos).

Fig 9A shows the differentially expressed cytokines by infection status for the irea.pos method. As shown, all the indicated differentially expressed cytokines (i.e., IL30, LIGHT and IL17D) are upregulated in the PBS condition relative to the infected categories. While the relation of these cytokines to the PBS control condition can be further investigated, we do not observe upregulation of any interferon-specific cytokines in the infected (MS-1/MS-2) categories like we did for the estimates generated using the MouSSE method.

Similar to the COVID19 analysis, we can investigate whether the application of the MouSSE signaling estimation strategy (i.e., weighting both positive and negative genes) can further improve the performance of gene sets constructed using the IREA method. Fig 9B shows the application of the irea.mousse method. As compared to Fig 9A, we observe upregulation of cytokines specific to the infected categories (i.e., IFN$\lambda$2). The increased interferon-specific signaling in cytokines differentially expressed in the infected categories compared to the PBS control highlights the potential benefit of MouSSE's gene set construction and scoring strategy. By prioritizing expression from both positively and negatively weighted gene sets, this approach may enhance the performance of existing pathway or cytokine activity estimation techniques.

The results from the remaining comparative methods (i.e., naive, receptor, product, niches, progeny and seurat) on the mouse lymph node ST data are included in the supplement (see S35–S40 Figs). Importantly, these analyses did not identify any significant associations between cytokine activity and infection status.

## 5. Discussion

We propose a novel method, MouSSE, for cytokine activity estimation using murine scRNA-seq or ST data. MouSSE leverages the Immune Dictionary cytokine stimulation data to construct weighted gene set signatures for 86 cytokines and, for a target dataset, uses these gene sets to estimate cell-level cytokine activity. Our MouSSE method can be leveraged in clinical settings to generate single-cell level cytokine activity estimates from patient transcriptomics data. These estimates can inform therapeutic strategies by identifying dominant cytokine-driven pathways in a patient's immune response. For example, in the context of severe COVID19, if MouSSE identifies elevated IFN-I activity in specific immune cell subsets,

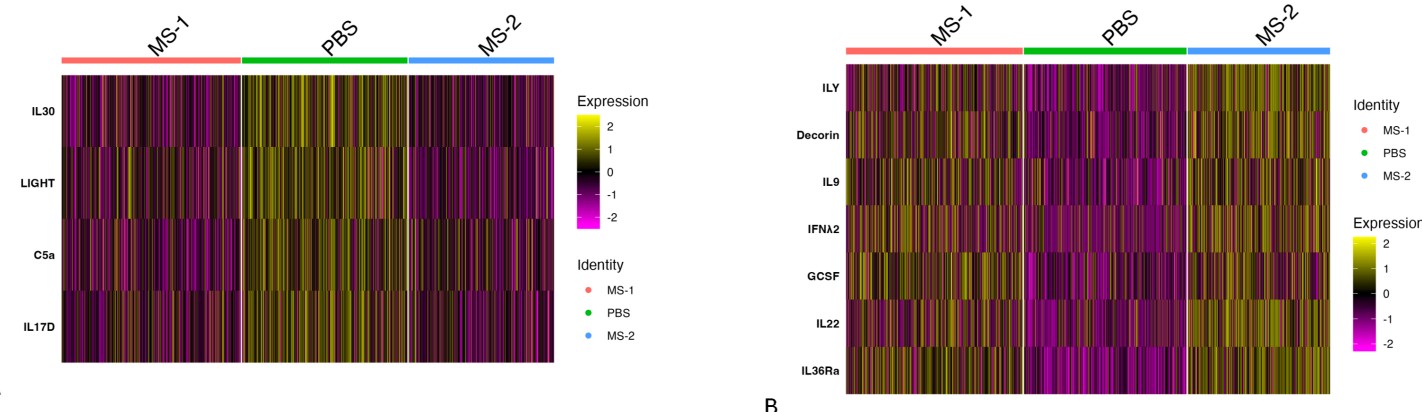

**Fig 9. Differentially expressed cytokine markers on mouse lymph node ST data consisting of 1092 spots by infection status (i.e., MS-1, MS-2 and PBS).** (Fig 9A) Heatmap of differentially expressed cytokines for the IREA-based gene sets scored using just positive weighted genes (irea.pos). (Fig 9B) Heatmap of differentially expressed cytokines for the IREA-based positive and negative weighted gene sets scored using MouSSE's weighting strategy (irea.mousse).

clinicians might consider targeting IFN-I signaling with JAK inhibitors [50] or other pathway-specific modulators. More broadly, this patient-level cytokine profiling could be used to stratify individuals by inflammatory risk or cytokine response profiles, enabling personalized immunomodulatory treatment plans. We provide the implementation of the MouSSE method, including the installation instructions and the associated vignette, via the MouSSE R package at https://github.com/azkajavaid/MousseR-package.

### 5.1. Limitations

A key strength of MouSSE compared to deep learning and optimal transport-based cell-cell communication methods is that the weighted gene sets used for scoring are interpretable and easy to customize. Furthermore, the gene set scoring strategy used by MouSSE, which prioritizes both positive and negative weighted genes, works well in practice and further improves cytokine activity estimation relative to methods that leverage expression from just positive weighted genes.

We note that correctly predicting the cytokine label corresponding to each sample via 5-fold cross-validation does not necessarily exactly correspond to the correct evaluation of the cytokine activity scores generated by MouSSE and comparative methods. One challenge with exactly evaluating the cytokine activity scores is that extensively documented ground truth data that provides the individual, mutually exclusive continuous activity scores for different cytokines does not currently exist. An exact comparison of cytokine activity estimates generated by MouSSE and comparative methods would entail conducting perturbation experiments where each cytokine is individually injected and tracked (i.e., labelled with a fluorescent antibody) to ensure to record its continuous cytokine activity before it potentially transforms into a different downstream cytokine. We are not currently aware of a database of experiment-based cytokine activity scores that would allow us to completely distinguish the activity of one cytokine against every other cytokine. We note this as a potential limitation of our manuscript and an inherent challenge in validating cytokine activity estimates in the absence of definitive ground truth data, although we believe the currently detailed evaluations still provide valuable insight into the relative performance of MouSSE.

For transparency, we included all available data in the heatmaps, even when the number of valid AUC-ROC scores for some methods was limited. We note that comparing AUC-ROC scores may be less meaningful when a method only produces valid AUC-ROC values for a small fraction of cytokines. However, our intent is to provide a comprehensive comparison across all cytokines for which AUC-ROC values can be computed.

### 5.2. Future directions

To further improve the performance of MouSSE, we can leverage reduced rank reconstruction. We have previously developed the SPECK method for unsupervised cell surface receptor abundance estimation using SVD-based reduced rank reconstruction. We plan to leverage our SPECK technique coupled with the gene set construction and gene set weighting strategy implemented in the MouSSE method to improve MouSSE's performance on very sparse scRNA-seq or ST datasets.

Our current implementation of MouSSE is cell-type agnostic. However, since cytokine signaling can be strongly cell-type-dependent, we plan to extend MouSSE in future work to perform cell-type-specific cytokine activity estimation. In addition, we note that an analysis of alternative five-fold splits (i.e., stimulation replicate condition) for cross-validation may further improve generalizability of MouSSE.

We note that, in this manuscript, we focus exclusively on mouse data and do not perform an application of the MouSSE method on non-mouse species, as this is beyond the scope of our current work. However, we acknowledge that extending the method to other species would require careful handling of differences in gene and cytokine nomenclature. In future work, we plan to incorporate support for cross-species applications, including robust gene ortholog mapping strategies.

## Supporting information

**S1 Fig. PR-AUC score for all 86 cytokines quantified using the MouSSE and comparative methods as computed on the 77,033 cell mouse lymph node scRNA-seq data.** Cytokine markers with an asterisk have the highest PR-AUC score when estimated using the MouSSE method (mousse).
(TIFF)

**S2 Fig. Specificity for all 86 cytokines quantified using the MouSSE and comparative methods as computed on the 77,033 cell mouse lymph node scRNA-seq data.** Cytokine markers with an asterisk have the highest specificity when estimated using the MouSSE method (mousse).
(TIFF)

**S3 Fig. Sensitivity for all 86 cytokines quantified using the MouSSE and comparative methods as computed on the 77,033 cell mouse lymph node scRNA-seq data.** Cytokine markers with an asterisk have the highest sensitivity when estimated using the MouSSE method (mousse).
(TIFF)

**S4 Fig. Precision for all 86 cytokines quantified using the MouSSE and comparative methods as computed on the 77,033 cell mouse lymph node scRNA-seq data.** Cytokine markers with an asterisk have the highest precision when estimated using the MouSSE method (mousse).
(TIFF)

**S5 Fig. NPV for all 86 cytokines quantified using the MouSSE and comparative methods as computed on the 77,033 cell mouse lymph node scRNA-seq data.** Cytokine markers with an asterisk have the highest NPV when estimated using the MouSSE method (mousse).
(TIFF)

**S6 Fig. Prevalence for all 86 cytokines quantified using the MouSSE and comparative methods as computed on the 77,033 cell mouse lymph node scRNA-seq data.** Cytokine markers with an asterisk have the highest prevalence when estimated using the MouSSE method (mousse).
(TIFF)

**S7 Fig. F1 score for all 86 cytokines quantified using the MouSSE and comparative methods as computed on the 77,033 cell mouse lymph node scRNA-seq data.** Cytokine markers with an asterisk have the highest F1 score when estimated using the MouSSE method (mousse).
(TIFF)

**S8 Fig. Detection rate for all 86 cytokines quantified using the MouSSE and comparative methods as computed on the 77,033 cell mouse lymph node scRNA-seq data.** Cytokine markers with an asterisk have the highest detection rate when estimated using the MouSSE method (mousse).
(TIFF)

**S9 Fig. Detection prevalence for all 86 cytokines quantified using the MouSSE and comparative methods as computed on the 77,033 cell mouse lymph node scRNA-seq data.** Cytokine markers with an asterisk have the highest detection prevalence when estimated using the MouSSE method (mousse).
(TIFF)

**S10 Fig. Balanced accuracy for all 86 cytokines quantified using the MouSSE and comparative methods as computed on the 77,033 cell mouse lymph node scRNA-seq data.** Cytokine markers with an asterisk have the highest balanced accuracy when estimated using the MouSSE method (mousse).
(TIFF)

**S11 Fig. PR-AUC comparison between every pair of cytokine for differential expression analysis trained to achieve distinguished signatures between pairs of cytokines as computed from cross-validated mouse lymph node target scRNA-seq data.** Rows represent cytokines to defined markers for (i.e., cytokines that are differentiated for).
(TIFF)

**S12 Fig. AUC comparison between every pair of cytokine for differential expression analysis trained to achieve distinguished signatures between pairs of cytokines as computed from cross-validated mouse lymph node target scRNA-seq data.** Rows represent cytokines to defined markers for (i.e., cytokines that are differentiated for).
(TIFF)

**S13 Fig. Specificity comparison between every pair of cytokine for differential expression analysis trained to achieve distinguished signatures between pairs of cytokines as computed from cross-validated mouse lymph node target scRNA-seq data.** Rows represent cytokines to defined markers for (i.e., cytokines that are differentiated for).
(TIFF)

**S14 Fig. Sensitivity comparison between every pair of cytokine for differential expression analysis trained to achieve distinguished signatures between pairs of cytokines as computed from cross-validated mouse lymph node target scRNA-seq data.** Rows represent cytokines to defined markers for (i.e., cytokines that are differentiated for).
(TIFF)

**S15 Fig. Precision comparison between every pair of cytokine for differential expression analysis trained to achieve distinguished signatures between pairs of cytokines as computed from cross-validated mouse lymph node target scRNA-seq data.** Rows represent cytokines to defined markers for (i.e., cytokines that are differentiated for).
(TIFF)

**S16 Fig. NPV comparison between every pair of cytokine for differential expression analysis trained to achieve distinguished signatures between pairs of cytokines as computed from cross-validated mouse lymph node target scRNA-seq data.** Rows represent cytokines to defined markers for (i.e., cytokines that are differentiated for).
(TIFF)

**S17 Fig. Prevalence comparison between every pair of cytokine for differential expression analysis trained to achieve distinguished signatures between pairs of cytokines as computed from cross-validated mouse lymph node target scRNA-seq data.** Rows represent cytokines to defined markers for (i.e., cytokines that are differentiated for).
(TIFF)

**S18 Fig. F1 score comparison between every pair of cytokine for differential expression analysis trained to achieve distinguished signatures between pairs of cytokines as computed from cross-validated mouse lymph node target scRNA-seq data.** Rows represent cytokines to defined markers for (i.e., cytokines that are differentiated for).
(TIFF)

**S19 Fig. Detection rate comparison between every pair of cytokine for differential expression analysis trained to achieve distinguished signatures between pairs of cytokines as computed from cross-validated mouse lymph node target scRNA-seq data.** Rows represent cytokines to defined markers for (i.e., cytokines that are differentiated for).
(TIFF)

**S20 Fig. Detection prevalence comparison between every pair of cytokine for differential expression analysis trained to achieve distinguished signatures between pairs of cytokines as computed from cross-validated mouse lymph node target scRNA-seq data.** Rows represent cytokines to defined markers for (i.e., cytokines that are differentiated for).
(TIFF)

**S21 Fig. Balanced accuracy comparison between every pair of cytokine for differential expression analysis trained to achieve distinguished signatures between pairs of cytokines as computed from cross-validated mouse lymph node target scRNA-seq data.** Rows represent cytokines to defined markers for (i.e., cytokines that are differentiated for).
(TIFF)

**S22 Fig. Proportion of cytokines with the highest AUC, PR.AUC, balanced accuracy, sensitivity, detection rate, F1 score, detection prevalence, negative predicted value, precision, specificity and prevalence across each of the ten methods for activity estimates generated using MouSSE scored with top 20 genes by absolute avg_log2FC as computed on cross-validated mouse lymph node target scRNA-seq data.**
(TIFF)

**S23 Fig. Proportion of cytokines with the highest AUC, PR.AUC, balanced accuracy, sensitivity, detection rate, F1 score, detection prevalence, negative predicted value, precision, specificity and prevalence across each of the ten methods for activity estimates generated using MouSSE scored with top 60 genes by absolute avg_log2FC as computed on cross-validated mouse lymph node target scRNA-seq data.**
(TIFF)

**S24 Fig. Proportion of cytokines with the highest AUC, PR.AUC, balanced accuracy, sensitivity, detection rate, F1 score, detection prevalence, negative predicted value, precision, specificity and prevalence across each of the ten methods for activity estimates generated using MouSSE scored with top 100 genes by absolute avg_log2FC as computed on cross-validated mouse lymph node target scRNA-seq data.**
(TIFF)

**S25 Fig. Proportion of cytokines with the highest AUC, PR.AUC, balanced accuracy, sensitivity, detection rate, F1 score, detection prevalence, negative predicted value, precision, specificity and prevalence across each of the ten methods for activity estimates generated using MouSSE scored with top 200 genes by absolute avg_log2FC as computed on cross-validated mouse lymph node target scRNA-seq data.**
(TIFF)

**S26 Fig. Heatmap of differentially expressed cytokines for mousse.pos method as constructed on COVID19 data.**
(TIFF)

**S27 Fig. Heatmap of differentially expressed cytokines for mousse.neg method as constructed on COVID19 data.**
(TIFF)

**S28 Fig. Heatmap of differentially expressed cytokines for irea.pos method as constructed on COVID19 data.**
(TIFF)

**S29 Fig. Heatmap of differentially expressed cytokines for irea.mousse method as constructed on COVID19 data.**
(TIFF)

**S30 Fig. Heatmap of differentially expressed cytokines for naive method as constructed on COVID19 data.**
(TIFF)

**S31 Fig. Heatmap of differentially expressed cytokines for receptor method as constructed on COVID19 data.**
(TIFF)

**S32 Fig. Heatmap of differentially expressed cytokines for product method as constructed on COVID19 data.**
(TIFF)

**S33 Fig. Heatmap of differentially expressed cytokines for Seurat method as constructed on COVID19 data.**
(TIFF)

**S34 Fig. Heatmap of differentially expressed cytokines for PROGENy method as constructed on COVID19 data.**
(TIFF)

**S35 Fig. Heatmap of differentially expressed cytokines for normalized ligand score (naive) as constructed on mouse lymph node ST data consisting of 1092 spots by infection status (i.e., MS-1, MS-2 and PBS).**
(TIFF)

**S36 Fig. Heatmap of differentially expressed cytokines for normalized receptor score (receptor) as constructed on mouse lymph node ST data consisting of 1092 spots by infection status (i.e., MS-1, MS-2 and PBS).**
(TIFF)

**S37 Fig. Heatmap of differentially expressed cytokines for ligand-receptor product score (product) as constructed on mouse lymph node ST data consisting of 1092 spots by infection status (i.e., MS-1, MS-2 and PBS).**
(TIFF)

**S38 Fig. Heatmap of differentially expressed cytokines for NICHES method (niches) as constructed on mouse lymph node ST data consisting of 1092 spots by infection status (i.e., MS-1, MS-2 and PBS).**
(TIFF)

**S39 Fig. Heatmap of differentially expressed cytokines for PROGENy method (progeny) as constructed on mouse lymph node ST data consisting of 1092 spots by infection status (i.e., MS-1, MS-2 and PBS).**
(TIFF)

**S40 Fig. Heatmap of differentially expressed cytokines for Seurat Perturbation Scores (seurat) as constructed on mouse lymph node ST data consisting of 1092 spots by infection status (i.e., MS-1, MS-2 and PBS).**
(TIFF)

## Acknowledgments

We would like to acknowledge the supportive environment at the Geisel School of Medicine at Dartmouth where this research was performed.

## Author contributions

**Conceptualization:** Azka Javaid, H. Robert Frost.

**Data curation:** Azka Javaid.

**Formal analysis:** Azka Javaid.

**Funding acquisition:** H. Robert Frost.

**Investigation:** Azka Javaid, H. Robert Frost.

**Methodology:** Azka Javaid, H. Robert Frost.

**Project administration:** Azka Javaid, H. Robert Frost.

**Resources:** H. Robert Frost.

**Software:** Azka Javaid.

**Supervision:** H. Robert Frost.

**Validation:** Azka Javaid.

**Visualization:** Azka Javaid.

**Writing – original draft:** Azka Javaid, H. Robert Frost.

**Writing – review & editing:** Azka Javaid, H. Robert Frost.

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
