## [Decision Letter · Decision Letter 0]

26 May 2025

PCOMPBIOL-D-25-00566

Mouse-Specific Single cell cytokine activity prediction and Estimation (MouSSE)

PLOS Computational Biology

Dear Dr. Javaid,

Thank you for submitting your manuscript to PLOS Computational Biology. After careful consideration, we feel that it has merit but does not fully meet PLOS Computational Biology's publication criteria as it currently stands. Therefore, we invite you to submit a revised version of the manuscript that addresses the points raised during the review process.

Please submit your revised manuscript within 60 days Jul 26 2025 11:59PM. If you will need more time than this to complete your revisions, please reply to this message or contact the journal office at ploscompbiol@plos.org. Please include the following items when submitting your revised manuscript:

We look forward to receiving your revised manuscript.

Kind regards,

Yesid Cuesta-Astroz, Ph.D

Academic Editor

PLOS Computational Biology

Mark Alber

Section Editor

PLOS Computational Biology

**Journal Requirements:**

At this stage, the following Authors/Authors require contributions: Azka Javaid, and H. Robert Frost. Please ensure that the full contributions of each author are acknowledged in the "Add/Edit/Remove Authors" section of our submission form.

5) We have noticed that you have uploaded Supporting Information files, but you have not included a list of legends. Please add a full list of legends for your Supporting Information files after the references list.

6) Thank you for stating "All datasets used in the manuscript are publicly accessible and referenced. Data include Liao et al. scRNA-seq dataset characterizing bronchoalveolar lavage fluid (BALF) immune cells from healthy individuals and patients with moderate and severe COVID19 and Lopez et al. ST dataset characterizing mouse lymph nodes treated with mycobacterium or PBS." Please update your Data Availability Statement to include the DOI/accession number of each dataset OR a direct link to access each dataset.

7) Please amend your detailed Financial Disclosure statement. This is published with the article. It must therefore be completed in full sentences and contain the exact wording you wish to be published.

8) Please provide a completed 'Competing Interests' statement, including any COIs declared by your co-authors. If you have no competing interests to declare, please state "The authors have declared that no competing interests exist.'

**Reviewers' comments:**

Reviewer's Responses to Questions

Reviewer #1: Javaid et al developed a method called MouSSE to predict mouse cytokine activities in single cell data or spatial transcriptome data. The method is based on large-scale perturbation scRNA-seq data for 86 murine cytokines. The method is evaluated by comparing with 9 other strategies. The method is applied to COVID single cell data and lymph node spatial transcriptomic data.

While it’s appreciated that the results are in details and the structure of the manuscript is clear, there are some flaws in the manuscript affecting full understanding of the story, especially about the methodology and the application range of the method. This manuscript could be further considered if the following comments are addressed.

1. References are not organized properly.

(1) The indexes in the main text are in a messy order. For example, the first reference is No.3 and the second one is No.20.

(2) Many sentences in the introduction section needs proper references while they are missing. Please check every sentence in the introduction section and add necessary references.

(3) Also, the reference format in the reference section needs to be corrected.

2. (Page 4, section 2.5) Usually k-fold cross validation means training with k-1 folds and validating on 1 fold. Please explain why the setting in this manuscript is the opposite.

3. (Page 7, figure 2)

(1) The ability to “correctly predict the cytokine label corresponding to each sample” is different from “correctly evaluate the cytokine activity (score)”. Being able to do the former could not prove the ability to do the latter. The authors are supposed to use a strategy that could directly compare the evaluation of relative cytokine activity.

(2) ROC or AUC is used to evaluate classification problems. Here if the prediction output is a score, it’s not proper to use AUC to evaluate to evaluate the performance. If the prediction is a multi-class problem, it’s better to notify if the AUC is from “1 vs rest” or between two specific classes.

(3) Here it’s better to directly show some scores or prediction output for each cytokine by these methods to compare relative ability of different methods. It’s not informative to show just a fraction of highest AUC counts. Having better performance than other methods may not mean a good method, especially when the evaluation approach is not suitable for other methods. The readers are supposed to be informed directly at least how well this method itself could do the prediction by quantification results.

4. (Page8, figure 3)

(1) There is a row named “PBS” in the heatmap, it’s confused if this refers to a cytokine or the buffer. It it’s the buffer for negative control, then it’s unclear why there is a row showing the result for the buffer.

(2) It looks like some AUC value is around 0.5 or even lower than 0.5, which means the evaluation is no good than a random guess. Just like question 5, it’s not meaningful to rank No.1 in the prediction score, if all methods have low AUC scores. It’s meaningful only when the method could have a good performance on the prediction.

(3) Some methods only have AUC for a small fraction of cytokines, in this case, it’s meaningless to declare that MouSSE have higher AUC than them. For example, the authors state that “Figure 2b shows that approximately 90% of cytokines (n=78) have the highest AUC score when estimated using the MouSSE method as compared to the IREA and Seurat-based strategies”, while the Seurat based method only have 4 valid AUC scores in the heatmap.

5. (Section 4.2)

(1) Applying the method on human COVID data and show that the trend for some specific cytokines/pathways meet previous reports in figure 4-7 is not a good way to validate that the method could be applied on human. A rigorous validation requires comparison of prediction of all cytokines with previous human data quantitatively and demonstration that the method out-performs all previous methods working on human data like section 4.1.

(2) The authors are also supposed to explain in more details how the method works on non-mouse species, when the training is done on mouse data. For example, the nomenclature of cytokine names could be different in different species and it’s unclear how the map from mouse gene names to other species is done. Also, it’s unclear whether this method could be directly used on more species beyond mouse and human, like other mammals or non-mammals.

6. It’s appreciated to see the authors include the download approach in GitHub. However, the authors are suggested to write the step-by-step protocol on how to run MouSSE directly in README file on GitHub. It’s not easy for researcher not familiar with R to find out how to run this package.

Reviewer #2: Strengths:

Novelty: MouSSE addresses key limitations of the authors’ prior method (SCAPE) by expanding cytokine coverage (86 vs. 41), using murine-specific scRNA-seq perturbation data, and incorporating bidirectional gene weights.

Technical Rigor: The stratified cross-validation on a large-scale dataset (385k cells) and comparisons against nine alternative methods (e.g., NICHES, PROGENy) are comprehensive.

Biological Relevance: Applications to COVID-19 and ST data yield results consistent with literature (e.g., IFN-I upregulation in severe COVID-19), showcasing practical utility.

Reproducibility: Code and datasets are publicly accessible, and the method is implemented as an R package.

Weaknesses and Concerns:

Parameter Justification: The selection of the top 60 genes per cytokine (based on log2 fold-change) is arbitrary. Sensitivity analyses or rationale for this threshold (e.g., stability across gene set sizes) are absent.

Biological Validation: The study relies solely on computational metrics (AUC, correlation with Hallmark pathways). Independent experimental validation (e.g., protein-level cytokine measurements) would strengthen claims.

Overfitting Risks: Gene sets are derived from the Immune Dictionary, which is also used for cross-validation. Testing on entirely independent datasets (beyond Liao and Lopez et al.) would better assess generalizability.

Cell-Type Heterogeneity: The cell-type-agnostic approach may obscure cell-specific signaling dynamics. The discussion acknowledges this limitation but does not quantify its impact on results.

Methodological Clarity: The modified VAM’s gamma distribution fitting and its contribution to scoring are under-explained. A pseudocode or schematic would improve transparency

How does MouSSE handle cytokines with overlapping gene signatures (e.g., IL-4/IL-13)? Does the differential expression analysis against all other cytokines sufficiently mitigate redundancy?

Why does the positive-weighted of MouSSE perform comparably to the full method in ST data but not in COVID-19 scRNA-seq? Is this due to dataset sparsity or biological context?

Include sensitivity analyses for key parameters (e.g., gene set size, fold-change thresholds).

Discuss limitations of AUC (e.g., class imbalance in cytokine labels) and supplement with precision-recall curves if applicable.

Expand the biological interpretation: How might MouSSE’s predictions inform therapeutic strategies (e.g., targeting IFN-I in severe COVID-19)?

Reviewer #3: MouSSE is a well-motivated attempt to quantify cytokine activity in scRNA-seq data by combining murine stimulation signatures with a variance-adjusted Mahalanobis (VAM) score. Benchmarking against nine competing tools and two external datasets shows promising average AUC gains. The idea is sound and potentially valuable, but the current manuscript and code require substantial revisions before the work can be considered robust and reproducible.

Major issues that must be addressed:

1. Incomplete capture of canonical biology (IL-6 in severe COVID-19): IL-6 is universally recognized as the hallmark cytokine of severe COVID-19, its blood concentrations correlate best with respiratory failure and mortality, and antagonists such as tocilizumab are among the few immunomodulators with proven clinical benefit. Any tool that claims to quantify cytokine activity in single-cell data should therefore detect a clear IL-6 signal in broncho-alveolar lavage from severe cases. Your current results do not highlight IL-6, which calls into question the biological validity of the murine signatures, the weighting scheme, or the averaging strategy. Resolving this discrepancy is essential before the approach can be considered reliable.

- Please verify that the IL-6 module is active, consider whether cell-type-restricted expression, single-cell dropout (secreted cytokine) or the murine training signatures could mask the signal, and explain or correct this discrepancy.

- Figure 4b suggests IL-6 is higher in the severe group, yet IL-6 is not discussed. What are the log₂ fold-change and FDR-adjusted P values for IL-6 under both the full MouSSE score and the positive-only variant?

- Are rows in the heatmaps filtered by statistical significance before plotting? Please specify the exact thresholds so readers can interpret colour gradients relative to inclusion in the narrative.

- The positive-only score appears to recover a canonical severe-COVID-19 cytokine that the full score attenuates, yet you conclude the signed-weight model is superior overall. How do you reconcile this single-case discrepancy with the aggregate AUC advantage you report?

- Could negative-weight genes derived from mouse lymph-node stimulation invert or dampen the IL-6 signal in human BALF? Would re-deriving the IL-6 signature from human in-vitro data mitigate this issue?

- Why is IL-6 not included in the Fig6 analysis?

2. Cross-validation strategy and performance evaluation:

- Current five-fold splits operate at the cell level. Splitting by stimulation replicate or donor would give a fairer estimate of generalizability. Provide results for such experiment-level hold-outs. Add an independent mouse dataset from a separate laboratory.

- Provide additional metrics beyond AUC, such as report PR-AUC, Spearman correlation with cytokine protein or phospho-STAT measurements, and computing resources (runtime and memory differences arise between MouSSE and the nine comparators).

3. NaN propagation in the public vignette: The PBMC3k example triggers NaNs produced warnings from sqrt(diag()) inside VAM. Diagnose the cause (are these zero variance genes), document how vamForCollection handles NaNs, and implement or describe pre-filters that prevent silent score loss.

4. Manuscript formatting and incorrect statements:

- No line numbers

- Numerous missing references (Introduction starts at ref 3, then jumps to 20).

- Introduction oversimplifies pro-inflammatory cytokines by omitting their pathogen-clearance role

- IFN-γ is not type I IFN

5. Data and code availability: Source datasets are public and the MouSSE R package is on GitHub, but the current vignette’s NaN warnings undermine trust. Resolve the warnings, document causes and mitigation and verify that all figure-generation scripts reproduce the manuscript results without manual tweaks. Consider uploading full pipelines, pre-computed gene sets to Zenodo to obtain doi-backed reference to ensure reproducibility.

**Have the authors made all data and (if applicable) computational code underlying the findings in their manuscript fully available?**

Reviewer #1: Yes

Reviewer #2: Yes

Reviewer #3: Yes

PLOS authors have the option to publish the peer review history of their article (what does this mean?). If published, this will include your full peer review and any attached files.

Reviewer #1: No

Reviewer #2: No

Reviewer #3: No

**Figure resubmission:**
---

## [Decision Letter · Decision Letter 1]

1 Sep 2025

Dear Dr. Javaid,

We are pleased to inform you that your manuscript 'Mouse-Specific Single cell cytokine activity prediction and Estimation (MouSSE)' has been provisionally accepted for publication in PLOS Computational Biology.

Best regards,

Yesid Cuesta-Astroz, Ph.D

Academic Editor

PLOS Computational Biology

Mark Alber

Section Editor

PLOS Computational Biology

Reviewer #1:

Reviewer #2:

Reviewer #3:

Reviewer's Responses to Questions

**Comments to the Authors:**

Reviewer #1: The authors have addressed all concerns in details. There is no more comment.

Reviewer #2: This manuscript introduces MouSSE, which builds weighted, bidirectional cytokine gene sets from the Immune Dictionary and scores them via a modified VAM to produce cell-level cytokine activity in murine scRNA-seq/ST data. Coverage (86 cytokines), extensive benchmarking (10 comparators, 5-fold CV; multiple metrics), and open R implementation are notable strengths. The revised version adds clarity, more metrics (including PR-AUC), sensitivity analyses, and resolves a NaN issue in the package. Note: If space allows, add one line stating that all figures are reproducible from scripts in the repo.

Reviewer #3: The authors have successfully and thoroughly addressed all of my questions and concerns. The revisions, particularly the corrected IL-6 analysis in the COVID-19 data and the expansion of performance metrics, have substantially improved the manuscript's rigor and biological validity.

Well done, and great work!

**Have the authors made all data and (if applicable) computational code underlying the findings in their manuscript fully available?**

Reviewer #1: Yes

Reviewer #2: Yes

Reviewer #3: Yes

PLOS authors have the option to publish the peer review history of their article (what does this mean?). If published, this will include your full peer review and any attached files.

Reviewer #1: No

Reviewer #2: No

Reviewer #3: No

---

## [Editor Report · Acceptance letter]

PCOMPBIOL-D-25-00566R1

Mouse-Specific Single cell cytokine activity prediction and Estimation (MouSSE)

Dear Dr Javaid,

I am pleased to inform you that your manuscript has been formally accepted for publication in PLOS Computational Biology. Your manuscript is now with our production department and you will be notified of the publication date in due course.

With kind regards,

Zsofia Freund
